# Identification of Targetable Liabilities in the Dynamic Metabolic Profile of *EGFR*-Mutant Lung Adenocarcinoma: Thinking beyond Genomics for Overcoming EGFR TKI Resistance

**DOI:** 10.3390/biomedicines10020277

**Published:** 2022-01-26

**Authors:** Anastasios Gkountakos, Giovanni Centonze, Emanuele Vita, Lorenzo Belluomini, Michele Milella, Emilio Bria, Massimo Milione, Aldo Scarpa, Michele Simbolo

**Affiliations:** 1ARC-NET Applied Research on Cancer Center, University of Verona, 37134 Verona, Italy; anastasios.gkountakos@univr.it (A.G.); aldo.scarpa@univr.it (A.S.); 21st Pathology Division, Department of Pathology and Laboratory Medicine, Fondazione IRCCS Istituto Nazionale dei Tumori, 20133 Milan, Italy; giovanni.centonze@istitutotumori.mi.it (G.C.); massimo.milione@istitutotumori.mi.it (M.M.); 3Comprehensive Cancer Center, Fondazione Policlinico Universitario Agostino Gemelli IRCCS, 00168 Rome, Italy; emanuele.vita@unicatt.it (E.V.); emilio.bria@unicatt.it (E.B.); 4Department of Medical Oncology, Università Cattolica del Sacro Cuore, 00168 Rome, Italy; 5Medical Oncology, Department of Medicine, University of Verona, 37134 Verona, Italy; lorenzo.belluomini@univr.it (L.B.); michele.milella@univr.it (M.M.); 6Department of Diagnostics and Public Health, University of Verona, 37134 Verona, Italy

**Keywords:** lung adenocarcinoma, epidermal growth factor receptor, tyrosine kinase inhibitor, resistance, metabolism

## Abstract

The use of epidermal growth factor receptor (EGFR) tyrosine kinase inhibitors (TKIs) as first-line treatment in patients with lung adenocarcinoma (LUAD) harboring EGFR-activating mutations has resulted in a dramatic improvement in the management of the disease. However, the long-term clinical benefit is inevitably compromised by multiple resistance mechanisms. Accumulating evidence suggests that metabolic landscape remodeling is one of the mechanisms that EGFR-mutant LUAD cells activate, thus acquiring higher plasticity, tolerating EGFR TKI-mediated cytotoxic stress, and sustaining their oncogenic phenotype. Several metabolic pathways are upregulated in EGFR TKI-resistant models modulating the levels of numerous metabolites such as lipids, carbohydrates, and metabolic enzymes which have been suggested as potential mediators of resistance to EGFR TKIs. Moreover, metabolites have been shown to carry signals and stimulate oncogenic pathways and tumor microenvironment (TME) components such as fibroblasts, facilitating resistance to EGFR TKIs in various ways. Interestingly, metabolic signatures could function as predictive biomarkers of EGFR TKI efficacy, accurately classifying patients with EGFR-mutant LUAD. In this review, we present the identified metabolic rewiring mechanisms and how these act either independently or in concert with epigenetic or TME elements to orchestrate EGFR TKI resistance. Moreover, we discuss potential nutrient dependencies that emerge, highlighting them as candidate druggable metabolic vulnerabilities with already approved drugs which, in combination with EGFR TKIs, might counteract the solid challenge of resistance, hopefully prolonging the clinical benefit.

## 1. Introduction

Lung cancer remains one of the most frequent and deadly types of cancer for both genders, while predictions about the future do not suggest dramatic improvements in incidence and mortality rates [1,2]. There are different histological types of primary lung cancer, classified mainly into non-small-cell lung cancer (NSCLC) and small-cell lung cancer, with an annual incidence of approximately 85% and 15%, respectively. Lung adenocarcinoma (LUAD) is the most common subtype of NSCLC, accounting for around 40% of NSCLC cases [3]. In 2004, the discovery of activating mutations in the gene encoding for epidermal growth factor receptor (EGFR) opened a new era for LUAD treatment. *EGFR* mutations occur in nearly 50% of Asian and approximately 15% of Caucasian patients with LUAD [4]. Specifically, exon 19 deletions at codons 746–750 and L858R mutation in exon 21 account for approximately 90% of *EGFR*-sensitizing mutations. Several phase 3 clinical trials have consistently shown the superior efficacy of first-generation (e.g., gefitinib, erlotinib, icotinib) and second-generation (e.g., afatinib) EGFR tyrosine kinase inhibitors (TKIs) in comparison with standard first-line platinum-based chemotherapy in patients with LUAD harboring these *EGFR*-activating mutations [5,6,7,8,9]. Despite the robust clinical activity exerted by EGFR TKIs, the majority of patients develop resistance after an average period of 9–15 months. Acquired *EGFR*-dependent and *EGFR*-independent genetic alterations are known to be associated with resistance to EGFR TKIs. Among them, the secondary “gatekeeper” mutation *EGFR* T790M in exon 20 is the dominant resistance mechanism occurring in around 50% of patients treated with first- or second-generation TKIs [10,11]. Osimertinib is an irreversible third-generation EGFR TKI that is highly selective for both *EGFR*-activating mutations and the *EGFR* T790 mutation. In the FLAURA trial, osimertinib resulted in improved survival outcomes compared to first-generation EGFR TKIs, and consequently entered in clinical practice as standard therapy for treatment-naïve patients [12,13]. Unfortunately, similarly to patients treated with previous-generation EGFR TKIs, patients inevitably develop secondary resistance to osimertinib. In this setting, tertiary acquired mutations such as the *EGFR* C797S mutation in exon 20 are reported in only 7% of patients treated with first-line osimertinib [14,15]. In contrast, multiple co-existing molecular alterations such as *KRAS* mutations, *MET* amplification, small-cell transformation, and gene fusions have been observed in a considerable percentage of patients [16,17]. The heterogeneity and coexistence of multiple molecular alterations pose a significant challenge due to the paucity of post-osimertinib pharmacological options and underline the need of new alternative approaches in order to overcome resistance.

Beyond genomics, accumulating evidence suggests that *EGFR*-mutant LUAD cells overcome EGFR TKI-mediated cytotoxicity through parallel resistance mechanisms, stemming from different multi-omics sources [18,19]. Cancer cells, in order to meet their high demands for energy, undergo a multifaceted metabolism rewiring [20]. In the early 20th century, physiologist Otto Heinrich Warburg described a metabolic process in cancer, where cancer cells not only increased glucose uptake but also switched their activity favoring the production of lactate regardless of oxygen availability—a phenomenon termed the Warburg effect. Today, altered tumor cell metabolism is critically involved in oncogenesis, metastasis, and resistance to treatment, and is generally recognized as a hallmark of cancer [21]. In this review, we discuss the effect of EGFR TKIs on downstream metabolism-related pathways in *EGFR*-mutant LUAD and how alterations in the levels of oncometabolites could induce resistance to therapy. Moreover, we describe promising metabolism-modulating interventions that might enhance EGFR TKI efficacy and combat the emergence of resistance.

## 2. *EGFR*-Mutant LUAD Global Metabolic Profile

Physiologically, EGFR activation stimulates different signaling pathways such as PI3K/Akt and MEK/Erk that are tightly linked downstream with the regulation of a highly diverse set of metabolic pathways [22,23,24]. In the context of *EGFR*-mutant LUAD, the aberrant overactivation of EGFR signaling axis leads to increased glucose uptake, excessive lactate formation, upregulation of the pentose phosphate pathway (PPP), and de novo pyrimidine biosynthesis in *EGFR*-mutant in vitro models [25]. Moreover, the decisive role of EGFR signaling in reprogramming the metabolic landscape of *EGFR*-mutant LUAD has been demonstrated by treating gefitinib/erlotinib-sensitive (HCC827, PC9) and -resistant (H1975) *EGFR*-mutant LUAD cells. In particular, the treatment abrogated the activity of crucial metabolic pathways including glycolysis and PPP, downregulating effectively intermediate metabolites such as fructose 1,6-bisphosphate, lactate, and dihydroxyacetone phosphate in EGFR TKI-sensitive cells compared to resistant cells. Interestingly, quantification of these oncometabolites and establishment of cut-offs in the clinic might serve as auxiliary prediction biomarkers to EGFR TKI efficacy in patients with *EGFR*-mutant LUAD [25].

## 3. EGFR TKI Resistance and Glycolysis/Lactate Metabolism

Glycolysis is a multi-level metabolic process assigned to numerous enzymes. Pyruvate dehydrogenase complex (PDH) is a multienzyme complex that catalyzes the oxidative phosphorylation of pyruvate towards acetyl coenzyme A (Acetyl-CoA). However, phosphorylation of PDH by pyruvate dehydrogenase kinase 1 (PDHK1) inactivates it, resulting in the overproduction of lactate. Interestingly, mRNA levels of PDHK1 along with genes of multiple glycolytic enzymes have been found to be significantly higher in NSCLC compared to normal tissue [26]. Today, numerous metabolism-modulating agents are available, and some of them seem to exert an anti-cancer effect (Table 1). In this case, the inhibition of PDHK1 by dichloroacetate (DCA), a PDHK antagonist, in *EGFR*-mutant LUAD cells, enhanced mitochondria-mediated pyruvate oxidation, thus decreasing lactate production. Intriguingly, decreased mitochondrial function has been correlated with an epithelial–mesenchymal transition (EMT) phenotype, a well-known resistance mechanism to EGFR TKIs [26,27]. Combination of DCA with EGFR TKIs (erlotinib and rociletinib, a third-generation EGFR TKI) abrogated the resistant phenotype in EGFR TKI-resistant LUAD cell lines. However, when cell lines were already resistant, the effect of DCA was less profound than on parental cells, highlighting the crucial role that timely intervention might have also in the clinical setting [26].

The tumor microenvironment (TME) is a mixture of active and dynamic components, including a plethora of metabolites that all interact to promote cancer cell survival and proliferation. In particular, the lactate secreted by EGFR TKI-resistant cells is engulfed by cancer-associated fibroblasts (CAFs), triggering the overproduction of hepatocyte growth factor (HGF) and the subsequent activation of MET signaling, suggesting the presence of a non-cell-autonomous metabolism-based mechanism of resistance to EGFR TKIs [28]. Erlotinib-resistant in vivo models were generated by subcutaneously injecting the *EGFR* exon 19 deletion-containing LUAD HCC827 cell line sensitive to EGFR TKIs. Interestingly, derived cell lines from these models were not resistant to erlotinib in vitro. The resistant phenotype was rescued when CAF-enriched conditioned media was cocultured. Importantly, monocarboxylate transporter 4 (MCT4), a lactate transporter and well-known readout of the Warburg effect, was found to be increased in resistant HCC827 cells compared to parental cells, confirming the metabolic rewiring towards aerobic glycolysis. Moreover, stromal HGF and MCT4 were upregulated in NSCLC samples progressed after gefitinib/erlotinib treatment, suggesting potential clinical relevance of the findings. Collectively, the implication of lactate metabolism and MET signaling in EGFR TKI resistance warrants additional research exploring the efficacy of combinatorial treatments with lactate/MET inhibitors and EGFR TKIs [28].

In another study, the exposure of *EGFR*-mutant HCC827 and PC9 cells to osimertinib and gefitinib respectively induced rapid cell death. However, a subpopulation of treated cells evaded the EGFR TKI-induced cytotoxic effect using an EGFR-independent resistance mechanism. Expression profile analysis of gefitinib-resistant PC9 clones identified the downregulation of genes related with glutamine metabolism and the tricarboxylic acid (TCA) cycle [29]. Further analysis of microRNA levels in osimertinib/gefitinib-resistant and parental cells displayed miR-147b as one of the most upregulated microRNAs in resistant clones. Overexpression of miR-147b in HCC827 cells significantly increased their resistance to both gefitinib and osimertinib, while the ablation of miR-147b in H1975 increased sensitivity to osimertinib by 166-fold. Experiments exploring the miR-147b-mediated resistance mechanisms involved the succinate dehydrogenase complex subunit D (SDHD), encoding for one of the subunits of the TCA cycle enzyme succinate dehydrogenase (SDH), as a direct target of miR-147b. Metabolic analysis in osimertinib-sensitive and -resistant H1975 cells revealed abrogated levels of different TCA cycle intermediate metabolites in H1975-resistant cells. Specifically, succinate and 2-oxoglutarate levels were elevated, while fumarate and malate levels were decreased. Interestingly, knock-out of miR-147b partially reversed the levels of these metabolites, suggesting that the miR-147b/SDHD axis regulates TCA cycle homeostasis, eventually inducing EGFR TKI resistance. The important role of SDH in resistance occurrence was also demonstrated after treating H1975 cells with dimethyl malonate, an SDH inhibitor, in the presence of osimertinib, where the results exhibited the increased resistance [29].

**Table 1 biomedicines-10-00277-t001:** Metabolism-modulating agents with efficacy against EGFR TKI-resistant LUAD preclinical models.

Compound	Mechanism of Action	Downstream Effect	Reference
Dichloroacetate	PDHK inhibitor	Induction of Acetyl-CoA formation	[26]
Atorvastatin	CAV-1/GLUT-3 inhibitor	Inhibition of glucose uptake and cholesterol synthesis	[30]
PF-429242	SREBP1 inhibitor	Inhibition of lipid synthesis	[31]
Orlistat	FASN inhibitor	Inhibition of EGFR palmitoylation	[32]
Epalrestat	AKR1B1 inhibitor	Inhibition of GSH synthesis	[33]
PiperlongumineAuranofin	ROS inducing agent	Induction of oxidative stress	[34]
Buthionine sulfoximine	GSH synthesis inhibitor	Inhibition of GSH synthesis	[34]
1,25D	VDR agonist	Inhibition of stemness phenotype	[35]
AZ12756122	FASN inhibitor	Inhibition of stemness phenotype	[36]

Abbreviations: PDHK, pyruvate dehydrogenase kinase; Acetyl-CoA, acetyl coenzyme A; CAV-1, caveolin-1; GLUT-3, glucose transporter-3; FASN, fatty acid synthase; EGFR, epidermal growth factor receptor; SREBP1, sterol regulatory-element-binding protein 1; AKR1B1, aldo-keto reductase family 1 member B1; GSH, glutathione; ROS, reactive oxygen species; 1,25D, 1,25-dihydroxyvitamin D3; VDR, vitamin D receptor.

## 4. EGFR TKI Resistance and Fatty Acid Metabolism

In the clinical setting, the presence of *EGFR*-sensitizing mutations does not guarantee an equal survival benefit in all patients with *EGFR*-mutant LUAD. Therefore, and besides the inevitable resistance mechanisms, the high variability of response rate must be taken into consideration, demonstrating the urgent need for identifying the factors responsible for this reality. To that end, one study analyzed the serum metabolic profile of 44 patients with *EGFR*-mutant LUAD treated with icotinib and divided them into equal groups of good and poor responders, considering progression-free survival longer or shorter than 11 months [37]. Among 80 metabolites, a set of seven lipid factors based on their differential expression was able to distinguish good from poor responders. Specifically, lysophosphatidylcholine 16:1, lysophosphatidylcholine 22:5-1, and phosphatidylethanolamine 18:2 were upregulated in good responders, while ceramide 36:1-3, ceramide 38:1-3, sphingomyelin 36:1-2, and sphingomyelin 42:2 were elevated in poor responders [37].

Many non-oncological drugs have been repurposed for anti-cancer effects [38]. For example, statins are a class of cholesterol-lowering drugs widely prescribed to patients with hyperlipidemia. Intriguingly, there is growing evidence that statins augment the efficacy of EGFR TKIs in a synergistic manner [39,40]. Interestingly, exposure to gefitinib or erlotinib has been found to significantly downregulate and elevate cholesterol levels in *EGFR*-mutant sensitive (HCC827, PC9) and resistant (H1975) LUAD cell lines, respectively. The use of statins such as atorvastatin (ATV) has been demonstrated to abrogate the function of caveolin-1 (CAV-1), a factor that regulates cellular cholesterol homeostasis. Indeed, ATV abrogated the expression of CAV-1 and decreased the survival of *EGFR*-mutant LUAD cells by upregulating apoptosis. Interestingly, a combination of ATV with gefitinib significantly enhanced cell death compared to ATV alone. However, the addition of mevalonate, an intermediate metabolite of cholesterol formation, restricted the tumor cell suppression and reinduced the CAV-1 expression in ATV-treated cells, demonstrating the tight link between cholesterol and CAV-1 [30]. Glucose transporters 1 (GLUT-1) and 3 (GLUT-3) are the most upregulated transporters in cancer [41], and GLUT-3 is activated through EMT in NSCLC [42]. CAV-1 upregulated the expression of GLUT-3, then formed a heterodimer only in PC9-resistant cells increasing glucose uptake, while the EGFR TKI-resistant cells presented a higher dependency on CAV-1/GLUT-3-mediated glucose uptake for their survival compared to sensitive cells. Similarly, in vivo analysis has demonstrated that ATV-treated resistant xenograft models had decreased cholesterol levels inducing downregulation of the CAV-1/GLUT-3 axis, abrogating the glucose uptake, and resulting in lower tumor volumes [30].

Another study identified a higher expression of low-density lipoprotein receptor (LDLR) mediated by sterol regulatory-element-binding protein 1 (SREBP1) in *EGFR*-mutant LUAD cells compared to wild type (wt). Combinatorial treatment of ATV and EGFR TKIs resulted in higher inhibition than single treatment, highlighting the importance of lipid biosynthesis in EGFR TKI resistance [43]. Generally, elevated rates of lipogenesis are a characteristic of numerous tumors while SREBP1-mediated constitutive lipogenesis is one of the main characteristics of EGFR TKI-resistant LUAD cells and might represent a promising druggable target for overcoming the resistance [44,45]. Moreover, osimertinib along with EGFR TKIs of all generations decreased the mature form of SREBP1 (mSREBP1), Acetyl-CoA carboxylase (ACC) and fatty acid synthase (FASN) through mTORC2 inhibition. Osimertinib has a strong effect on the lipid profile of the *EGFR*-mutant LUAD cells by significantly reducing the presence of lipid droplets. Moreover, the lipidomic profile of osimertinib-treated cells has revealed decreased concentrations of lipid classes such as diacylglycerol, triacylglycerol, ceramides, and phosphatidylethanolamine [31]. Mechanistically, osimertinib induced the degradation of mSREBP1 by releasing the GSK3/FBXW7 axis. EGFR TKI-resistant cell lines had higher levels of mSREBP1, ACC, and FASN compared to parental cells, confirming the critical role of de novo lipid biosynthesis in EGFR TKI resistance. Similarly, the SREBP1/ACC/FASN axis was elevated in EGFR TKI-relapsed patients compared to tissues at baseline. Interestingly, patients who did not respond to treatment exhibited higher FASN expression levels than those who showed a response, indicating that the axis is an innate resistance mechanism. Genetic perturbation of SREBP1 reversed the resistance to osimertinib both in vitro and in vivo, while ectopic expression of SREBP1 rendered the otherwise EGFR TKI-sensitive cells more resistant to osimertinib [31]. Similar findings were also reported after using commercially available SREBP1 inhibitors, where a synergistic combination was revealed. The combination of osimertinib with PF-429242, a SREBP1 inhibitor, also induced a strong effect in xenograft in vivo models, downregulating the axis and eventually the lipid metabolism. Collectively, the findings clearly suggest that fatty acid metabolism is tightly associated with osimertinib resistance, and timely intervention could delay the onset of resistance [31]. Moreover, FASN-mediated palmitoylation (addition of 16-C saturated fatty acid palmitate) of EGFR sustained constitutive EGFR signaling promoting EGFR TKI resistance. Pharmacological inhibition of FASN using orlistat, an FDA-approved anti-obesity agent, led to the ubiquitin-mediated degradation of EGFR, increasing cell death in gefitinib-resistant in vitro and in vivo models [32].

In clinical routine, limited availability of tissue frequently does not allow molecular analysis or leads to inconclusive histological diagnosis [46]. Therefore, different sources of material that could surpass the tissue limitations should be explored. Pleural effusion (PE) is a common complication of lung cancer. Analysis of the lipidomic profile of PE samples from *EGFR*-mutant and *EGFR*-wt patients with LUAD was able to discriminate the presence of *EGFR* mutation based on a seven-lipid profile including mainly 20 and 22 carbon chain length polyunsaturated fatty acids (PUFAs) and phospholipids [47].

## 5. EGFR TKI Resistance and Redox Homeostasis

The cells during physiological processes such as oxidative metabolism or response to cytokines or pathogen infection produce by-products of so-called reactive oxygen species (ROS). When ROS are accumulated without clearance, they induce cell death [48]. Moreover, *EGFR*-mutant LUAD enhances glycolysis to support cell survival. At a specific stage of glycolysis, LUAD cells exert an activating transcription factor 4-mediated metabolic rewiring, shifting glycolysis towards serine synthesis as demonstrated by upregulation of serine synthesis pathway metabolic enzymes such as phosphoserine phosphatase and phosphoglycerate dehydrogenase. Serine is a precursor of cystine which leads to glutathione (GSH) formation, which acts as an antioxidant and participates in ROS scavenging, inhibiting cell death [49]. Another study performed RNA sequencing analysis and immunoblotting in EGFR TKI-resistant HCC827 and H1975 models, identifying the upregulation of aldo-keto reductase family 1 member B1 (AKR1B1). Downregulation of AKR1B1 restored the sensitivity to osimertinib both in vitro and in vivo, while its ectopic overexpression rescued the resistant phenotype, revealing a crucial role of AKR1B1 in the occurrence of resistance [33]. Transcriptomic and metabolic comparisons of the models identified GSH metabolism as one of the most altered pathways. Specifically, resistant cells demonstrated an upregulation of major metabolites in the GSH de novo synthesis pathway, a finding which was confirmed both in vivo and in relapsed patients who had higher levels of GSH and oxidative GSH in blood compared to EGFR TKI-sensitive patients. Mechanistical studies demonstrated that AKR1B1 promotes the cystine uptake and consequent GSH de novo synthesis by interacting with and activating p-STAT3, which in turn activates cystine transporter SLC7A11/xCT. The elevated levels of GSH act as a major antioxidant, scavenging ROS and protecting tumor cells from treatment-induced stress. Disruption of AKR1B1, either genetically or due to epalrestat, an approved antidiabetic agent, inhibited the increase of GSH. The combination of epalrestat and either gefitinib or osimertinib therefore resulted in the restriction of tumor growth and delayed relapse by abrogating the axis of AKR1B1/p-STAT3/SLC7A11 in resistant models [33].

An example of multi-omics interconnection in the EGFR TKI resistance involves branched-chain amino acid (BCAA) metabolism. BCAAs are a group of three essential amino acids: leucine, isoleucine, and valine. Alterations in BCAA metabolism in response to erlotinib treatment have been demonstrated in *EGFR*-mutant HCC827 cells [50]. Branched-chain amino acid aminotransferase 1 (BCAT1) is a BCAA-catabolizing enzyme that was significantly upregulated in gefitinib/erlotinib-resistant in vitro and in vivo models. Conversely, H3K9 methylation on the *BCAT1* promoter was gradually decreased during resistance acquisition, indicating a possible negative correlation between them. Aberrant BCAT1 activation led to a metabolic rewiring involving the redox pathway and increasing GSH generation, resulting in ROS scavenging and EGFR TKI resistance. Interestingly, BCAT1 and H3K9 were found to be negatively correlated in *EGFR*-mutant LUAD samples at baseline and relapse. Moreover, BCAT1 was higher in the relapse group and was correlated with poor response to treatment. The use of ROS-inducing agents such as piperlongumine or auranofin as well as buthionine sulfoximine, a GSH synthesis inhibitor, managed to partially overcome the gefitinib resistance both in vitro and in vivo [34]. Interestingly, redox signaling also promoted EGFR TKI resistance through other mechanisms such as the NRF2/ALDH1A1/GPX4-SOD2 axis, which regulates ROS metabolism [51,52]. A summary of the aforementioned metabolic-related resistance mechanisms is illustrated in Figure 1. Finally, another study highlighted the important differences in amino acid metabolism between naïve and EGFR TKI-resistant patients with *EGFR*-mutant LUAD. Specifically, PE-derived metabolites were analyzed between naïve and first-generation EGFR TKI-treated groups. Principal component analysis identified 34 metabolites that were differentially expressed between the groups, and specifically higher in the treatment group. Further classification shared these metabolites into six metabolic pathways associated with amino acid metabolism (phenylalanine, glycine, serine, threonine) and biosynthesis (phenylalanine, tyrosine, and tryptophan) [53].

## 6. EGFR TKI Resistance and Metabolism-Mediated Stemness

The acquisition of a stem-cell-like phenotype is among the resistance mechanisms that *EGFR*-mutant LUAD cells develop to tolerate EGFR TKI treatment [54]. Indeed, Iroquois-class homeodomain protein 4 (IRX4), a factor that positively regulates pluripotency transcription factors such as NANOG, Sox2, and CD133, providing cells with stemness characteristics, was found to be significantly higher in PC9 gefitinib-resistant cells than PC9 parental cells [35]. Of note is the use of 1,25-dihydroxyvitamin D3 (1,25D), the biologically active form of vitamin D which activates the vitamin D receptor (VDR), which has been reported to abrogate a stemness phenotype in ovarian cancer cells [55]. Similarly, treating PC9-resistant cells with 1,25D increased VDR expression while significantly diminishing the levels of IRX4 and NANOG by negatively regulating the TGF-β1/SMAD3/IRX4/NANOG axis. Interestingly, a combination of 1,25D with gefitinib resulted in significantly higher efficacy of treatment both in in vitro and in vivo gefitinib-resistant models compared to single treatment. Collectively, impairing stem-like characteristics in EGFR TKI-resistant cells, as in this case by activating the 1,25D/VDR axis, led to resistance mitigation [35]. Moreover, preliminary findings suggested that a combination of osimertinib with a novel FASN inhibitor, AZ12756122, presented a synergistic effect overcoming osimertinib resistance in PC9 cells, likely by attenuating cancer stem cell features [36].

## 7. Conclusions

Today, the clinical treatment decisions in *EGFR*-mutant LUAD are primarily based on tumor genetic profiles. However, patients with identical EGFR TKI-sensitizing alterations present various response rates and relapse-free survival. Therefore, the effective stratification of patients with *EGFR*-mutant LUAD towards a long-term clinical benefit remains an unmet challenge and suggests that additional mechanisms beyond genetic alterations might exist which co-determine patients’ treatment efficacy. Metabolic reprogramming is inevitably triggered by constitutively active EGFR signaling pathways, empowering cancer initiation and progression. As reviewed here, EGFR TKIs acutely affect numerous metabolic processes, predominantly glucose metabolism, fatty acid biogenesis, and redox homeostasis, increasing cell death and tumor reduction. However, it seems that these oncometabolites rapidly recover their levels, reactivating EGFR signaling and establishing a resistant phenotype. Here, we discussed different comprehensive studies which have used a wide range of metabolic-modulating agents with remarkable efficacy against EGFR TKI resistance. We also discussed metabolic vulnerabilities that could be addressed by already FDA-approved drugs used for other diseases such as hyperlipidemia and diabetes, thus accelerating their introduction into clinical use for EGFR TKI-treated patients. A serious concern about metabolism-modulating drugs is the systemic toxicity that might be triggered when they are delivered to unselected cancer patients lacking indication of metabolic dependency. However, experiments in *EGFR*-mutant LUAD in vivo models with metabolic agents summarized in this review showed a high safety profile without affecting vital points of the in vivo models such as body weight.

Liquid biopsy represents a promising alternative solution to traditional tissue analysis. Interestingly, one of the few FDA-approved applications of liquid biopsy is the Cobas *EGFR* Mutation Test v2 as a companion diagnostic test for EGFR TKI-eligible NSCLC patients. Although the findings are limited, biofluids such as serum and PE of patients with *EGFR*-mutant LUAD could be used as sources of metabolites and provide information regarding treatment efficacy. The regular diet of patients naturally shapes the availability of nutrients. Although no dietary patterns such as an alkaline diet have been introduced in the treatment management of patients with lung cancer, future clinical trials investigating whether patient diet affects EGFR TKI response would be of high interest. Moreover, cancer-related cachexia is a catabolic syndrome that affects an important percentage of patients with NSCLC, and has been associated with patient survival and treatment efficacy. Therefore, future approvals of metabolism-targeted therapies should take into serious consideration this aspect, since drugs that block different anabolic processes might further aggravate the disease.

Finally, as presented here, in the presence of *EGFR*-sensitizing alterations, metabolomics and epigenomics collaborate to promote EGFR TKI resistance in *EGFR*-mutant LUAD. Although many -omics regarding EGFR TKI resistance in *EGFR*-mutant LUAD are still understudied, preliminary insights indicate that resistance occurrence is mediated at a multi-omics level and evaluation of EGFR TKI treatment should be performed with a more holistic approach. Collectively, we highly encourage the continuation of metabolism-oriented studies in *EGFR*-mutant LUAD for the identification of predictive and prognostic metabolic biomarkers. Moreover, future research should include clinical studies enrolling patients with *EGFR*-mutant LUAD and exploring the potential clinical value of translational insights from laboratory-based findings regarding the metabolism-targeted therapies in combination with EGFR TKIs.

## Figures and Tables

**Figure 1 biomedicines-10-00277-f001:**
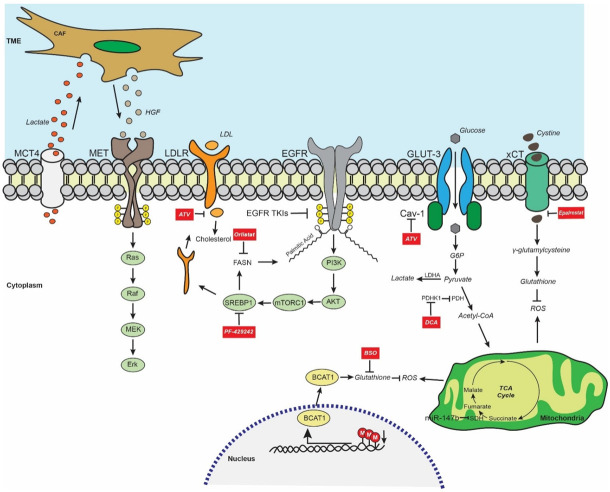
Overview of the metabolic remodeling in *EGFR*-mutant LUAD and the proposed treatment strategies for circumventing the EGFR TKI resistance. EGFR TKI-resistant LUAD cells secrete lactate, which is incorporated by CAFs located in TME, leading to overproduction of HGF ligands and subsequent activation of MET signaling and its downstream oncogenic effectors such as Ras/Raf/MEK/Erk signaling pathway. Moreover, the FASN-mediated EGFR palmitoylation allows the maintenance of EGFR constitutive signaling in EGFR TKI-resistant cells. Abrogation of palmitic acid synthesis using the anti-obesity drug orlistat (FASN inhibitor) promotes EGFR degradation. The use of statins such as ATV significantly downregulates the cholesterol levels, decreasing glucose uptake and lipogenesis simultaneously in *EGFR*-mutant LUAD. Further restriction of fatty acid synthesis by inhibiting SREBP1 using PF-429242 impairs both the levels of FASN and LDLR. Interestingly, blocking the PDHK1-mediated phosphorylation of PDH by treating *EGFR*-mutant LUAD cells with DCA shifts the glucose metabolism towards oxidative phosphorylation, preventing the accumulation of lactate, which has been correlated with EGFR TKI resistance. Furthermore, EGFR TKI-resistant cells have elevated levels of glutathione, the master antioxidant, thus efficiently scavenging ROS, escaping oxidative stress and cell death. Treatment with drugs such as epalrestat and BSO reduces glutathione synthesis, allowing ROS accumulation which eventually leads to ROS-mediated cell death in EGFR TKI-resistant models. Finally, epigenetic downregulation of SDH activity by miR-147b induces TCA cycle arrest and increased levels of succinate in EGFR TKI-tolerant cells. Abbreviations: CAF, cancer-associated fibroblast; TME, tumor microenvironment; MCT4, monocarboxylate 4; HGF, hepatocyte growth factor; LDLR, low-density lipoprotein receptor; LDL, low-density lipoprotein; EGFR, epidermal growth factor receptor; GLUT-3, glucose transporter 3; Cav-1, caveolin-1; G6P, glucose 6-phosphate; TCA cycle, tricarboxylic acid cycle; SDH, succinate dehydrogenase; ATV, atorvastatin; FASN, fatty acid synthase; PI3K, phosphoinositide 3-kinase; mTORC1, mTOR complex 1; ROS, reactive oxygen species; DCA; dichloroacetate; PDH, pyruvate dehydrogenase; PDHK1, pyruvate dehydrogenase kinase 1; Acetyl-CoA, acetyl coenzyme A; LDHA, lactate dehydrogenase A; SRBP1, sterol regulatory-element-binding protein 1; BSO, buthionine sulfoximine; BCAT1, branched-chain amino acid aminotransferase 1.

## Data Availability

Not applicable.

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
