# Peer review of "Identification of Targetable Liabilities in the Dynamic Metabolic Profile of EGFR-Mutant Lung Adenocarcinoma: Thinking beyond Genomics for Overcoming EGFR TKI Resistance"

_biomedicines, 2022, doi:10.3390/biomedicines10020277_

Round 1
Reviewer 1 Report
This review reports on metabolic mechanisms that may mediate resistance to EGFR-targeting TKIs. Overall, I thought this review was very well-written and I do not have further suggestions for improvement.
Author Response
On behalf of all the authors, I would like to thank you very much for the revision of the manuscript and your kind comments.
Reviewer 2 Report
Remarks about the manuscript:
The authors have presented an excellent review on "Identification of targetable liabilities in the dynamic metabolic profile of EGFR-mutant lung adenocarcinoma: Thinking beyond genomics for overcoming EGFR TKI resistance." It is an exciting review manuscript with excellent information related to EGFR-TKI treatment and their possible synergistic effect with metabolic factors. This manuscript highlighted many rational points with relevant references to overcome the limitation of EGFR-TKI treatment.
I thoroughly enjoyed reading and reviewing it, especially the lipids and fatty acid metabolism-related studies. The current manuscript also pointed out future studies about the combination of EGFR-TKI treatment with other FDA-approved drugs molecules for improved treatment of Lung cancer disease.
This review article presents up-to-date knowledge and biomedical applications of EGFR-TKI with current clinical trials and updated results. This review article contains proper references with related descriptions and subsequent applications.
In my view, this manuscript (ID: Biomedicines-1508670) can be accepted in the present form: The English language of this manuscript is very smooth with clarity.
Author Response

(The authors gave the same response as above.)

Reviewer 3 Report
The manuscript "Identification of targetable labilities in the dynamic metabolic profile of EGFR-mutant lung adenocarcinoma: Thinking beyond genomics for overcoming EGFR TKI resistance" written by Gkountakos A, Centonze G, Vita E, Belluomini L, Milella M, Bria E, Milione M, Scarpa A and Simbolo M, is a review describing metabolic changes in EGFR-mutated lung adenocarcinoma. These tumors, treated with tyrosine kinase inhibitors (TKI), often become resistant to them. The authors analyze numerous pathways in glucose and fatty acid metabolism, as well as in maintenance of redox homeostasis, which could be targeted with metabolism-modulating agents to fight EGFR –TKI resistance.
The manuscript is well-written, concentrated on the topic which is presented with numerous data, in logic order. The references cited are up-to date.
Minor comments:
-de novo should be written in italics
page 3: 2-oxoglutarate
page 5, second paragraph: when ROS... they induce cell death.
page 7: font and position of abbreviations belonging to Figure 1
Conclusions: A serious concern.... that might be triggered
font of References
Author Response
On behalf of all the authors, I would like to thank you very much for the detailed revision of the manuscript and your kind comments. Please find bellow the modifications according your suggestions: -de novo should be written in italics, changed-page 3: 2-oxoglutarate, changed
-page 5, second paragraph: when ROS... they induce cell death, changed
-page 7: font and position of abbreviations belonging to Figure 1
-Conclusions: A serious concern.... that might be triggered, changed
- font of References, changed according journal's requirements